# Assessment of Financial Development of Countries Based on the Matrix of Financial Assets

Galina Gospodarchuk [1,*] and Elena Zeleneva [2]

1 Department of Finance and Credit, National Research Lobachevsky State University of Nizhny Novgorod, 603022 Nizhny Novgorod, Russia
2 Department of Banking and Financial Markets, Financial University under the Government of the Russian Federation, 125167 Moscow, Russia; eszeleneva@fa.ru
* Correspondence: gospodarchukgg@iee.unn.ru

**Abstract:** Building an adequate system of indicators to assess the financial development of countries and its practical application can improve the robustness and effectiveness of government decision-making. This paper aims to create such a system. The study used the methods of structured system, comparative, matrix, and gap analysis. The key outcome of the study is a matrix system of indicators for assessing the financial development of countries. This indicator system is based on a matrix of all financial assets. Elements of the matrix of financial assets were calculated in relation to the population and used as indicators of the level of financial development of countries as a whole and in the context of financial instruments and sectors of the economy. Simultaneous recording of financial assets across the entire range of financial instruments and sectors of the economy, as well as their interrelations, is a relatively new direction for financial development assessment. The study produced criteria for the qualitative assessment of the level of the financial development of countries. Testing of the developed matrix system of indicators and criteria for financial development was carried out on current and potential members of OECD (OECD+) for the period 2018–2019. As part of the testing, the level of financial development of the analyzed countries was calculated, their ranking was gauged, and international positions were determined. A structural analysis of the financial development of OECD+ countries in terms of types of financial assets (instruments) and sectors of the economy was carried out. Promising areas of Italy's financial development have been identified. The test results confirmed that the matrix system of indicators and the developed criteria are an objective and convenient tool for assessing the level of financial development of countries. Their use makes it possible to increase the complexity and quality of the analysis of financial development, and it also forms a platform for making evidence-based and effective decisions in the development of national strategic documents.

**Keywords:** financial development; financial development indicators; financial development analysis; financial development strategy



## 1. Introduction

In recent years, there has been a significant increase in the number of strategic documents aimed at the financial development of countries. These include Corporate Plan 2021–2025 (ASIC 2021), Strategic Planning 2013–2023 (SEC 2013), Business Plan 2021–2022 (FCA 2021), Digitalization Strategy 2018–2023 ((Federal Financial Supervisory Authority 2018), Strategic Plan 2020–2024 (DG FISMA 2020), National Strategy for Financial Inclusion 2019–2024 (RBI 2019), Priorities for 2022 (IIROC 2021), Strategy 2020–2022 (Dutch Authority for the Financial Markets 2020), Finanstilsynet Strategy 2019–2022 (Financial Supervisory Authority of Norway 2019), Business Plan 2021–2022 (DFSA 2021), Strategic Plan 2018–2022 (SEC 2018), and Strategic goals 2021–2024 (FINMA 2021)). These provide key directions for the development of financial markets for 2022–2024 (Bank of Russia 2021). This is due

to the fact that financial development has a significant impact on the development of the economy. This is confirmed by the results of scientific research aimed at identifying the relationship between financial and economic development. In particular, the results of scientific studies carried out by Sehrawat and Giri (2015), Wait et al. (2017), Rekunenko et al. (2019), Sumarni (2019), Muhammad et al. (2021), and Setiawan et al. (2021) showed that financial development noticeably affects the restructuring of the economy, and makes a significant contribution to long-term economic growth and the functioning of relevant institutions.

According to scientific findings reflected in other publications, financial development is of great importance for the process of increasing cashflows (Kreso and Begovic 2013), expanding access to capital and financial services, and thereby creating conditions for the growth of investment and innovation activity of economic entities (Kar and Ozsahin 2016; Sayilir et al. 2018; Sinha and Shastri 2021). Financial development improves the efficiency of resource markets (H. U. R. Khan et al. 2019), boosts entrepreneurial activity (Dutta and Meierrieks 2021), reduces the size of the "grey" economy (Berdiev and Saunoris 2016; Canh and Thanh 2020), and reduces income inequality and poverty (Jung and Vijverberg 2019; Gnangnon 2021). Moreover, as noted by Lim (2017), Sayilir et al. (2018), Wang et al. (2018), Naceur et al. (2019), and Canh and Thanh (2021), financial development strengthens the financial stability of countries, since countries with more developed financial markets tend to have lower volatility.

A significant increase in the number of strategic documents containing the goals, objectives, and main directions of the financial development of countries drew the attention of scholars to the problem of measuring this development. This problem has emerged due to the fact that, so far, there has been no consensus in the literature on the best system of indicators that can fully reflect the scale of financial development of countries (Gnangnon 2021). Most studies examining the relationship between financial development and economic growth use selected financial development measures that, according to Sehrawat and Giri (2015), Cave et al. (2020), M. A. Khan et al. (2020), and Moyo and Le Roux (2021), often produce conflicting results. This is due to the fact that financial development is difficult to measure, since it is a broad concept that has several dimensions; therefore, conclusions drawn through individual proxies may lead to an incomplete understanding of the relationship between financial development and economic growth. Differences in conclusions make it difficult to assess the overall level of financial development and hinder the development of effective public policy.

The aim of the study is to develop a new system of indicators and criteria for measuring the level of financial development of countries, which make it possible to assess the scale of financial development (quantitative assessment), range countries by levels of development (qualitative assessment), and identify priority areas for this development.

This study consists of several sections. The first section includes an overview of current research related to the measurement of financial development. The second section is a presentation of a new concept for measuring the financial development of countries, an algorithm for calculating a universal indicator for measuring the scale of financial development, a description of methods for comparative and structural analysis of financial development, as well as criteria that allow determining priority areas for financial development in the medium and long term. The third section is devoted to testing the developed system of indicators in relation to current and potential members of the OECD countries (OECD+) and analyzing the results. The next section contains a discussion of the results of the study. The last section presents the main findings.

## 2. Literature Review

The analysis of scientific publications shows that indicators of the financial development of countries are formed based on two methodological approaches.

The first approach is based on quantitative indicators and includes two concepts. The foundation of the first concept comprises the traditional indicators of the monetary market:

ratio of M2 (Money Supply) to GDP (Gross Domestic Product), %; ratio of domestic credit provided by the financial sector to GDP, %; real interest rate, %; ratio of insurance and financial services to GDP, %; ratio of net inflow of foreign direct investment to GDP, %; ratio of total value of traded shares to GDP, %; stock market turnover ratio, %; ratio of deposits to GDP, %; ratio of loans to GDP, %. This concept is reflected in the publications of Berdiev and Saunoris (2016), Akel and Torun (2017), Wang et al. (2018), H. U. R. Khan et al. (2019), Jung and Vijverberg (2019), Sethi et al. (2020), Ayowole and Kalmaz (2020), Kim (2021), Moyo and Le Roux (2021), and Kandil et al. (2015).

The advantage of this concept is the availability of a large array of statistical data on the countries of the world that are in the public domain. These data could be used to build time series over a long period and apply them to analyze the impact of financial development on macroeconomic indicators. At the same time, this approach has its disadvantages. Conventional measures of financial development (ratio of M2 to GDP, %, ratio of domestic credit to GDP, %, etc.), which do not go beyond the assessment of savings mobilization, are not sufficient on their own to assess the financial system, and cannot be used to determine the right policy regarding the development of the financial sector and its impact on the economy.

The second concept uses a structured system of indicators. This system is based on the identification of the concepts of "financial development" and "development of the financial sector". At the same time, the development of the financial sector is understood as the development of financial markets and the sector of financial organizations. Thus, this system of indicators makes it possible to measure both the level of development of financial markets and the level of development of the sector of financial institutions. An analysis of publications shows that this concept is supported by Kreso and Begovic (2013), Luo et al. (2016), Jankovic and Gligoric (2017), Verma Gakhar and Kundlia (2018), Yadav et al. (2019), Rekunenko et al. (2019), Cave et al. (2020), Jiang et al. (2020), Abdul Karim et al. (2021), Muhammad et al. (2021), Trinugroho et al. (2021), Mignamissi and Djijo (2021), Sinha and Shastri (2021), and others.

Within this concept, the level of development of financial markets is measured using indicators of the development of the stock market and its segments, similar to the traditional indicators used in the first concept (Table 1).

**Table 1.** Indicators of the level of financial development of countries.

| Financial Market Development Indicators | Financial Sector Development Indicators |
|---|---|
| 1. Ratio of domestic credit provided by the financial sector to GDP, % <br> 2. Ratio of total value of traded shares to GDP, % <br> 3. Ratio of stock market capitalization to the volume of financial assets, % | 1. Ratio of financial sector assets to GDP, % <br> 2. Ratio of employees in the financial sector to the total number of employees, % <br> 3. Ratio of loans to deposits, % <br> 4. Ratio of deposits and loans to GDP, % |

Source: authors' research based on Kreso and Begovic (2013); Luo et al. (2016); Jankovic and Gligoric (2017); Verma Gakhar and Kundlia (2018); Yadav et al. (2019); Rekunenko et al. (2019); Cave et al. (2020); Jiang et al. (2020); Abdul Karim et al. (2021); Muhammad et al. (2021); Trinugroho et al. (2021); Mignamissi and Djijo (2021); and Sinha and Shastri (2021).

At the same time, the level of development of the sector of financial institutions is measured mainly by indicators of the development of the banking sector. This is due to the availability of publicly available statistical data. It is important to note that the indicators given in Table 1 are primary. As a rule, they are used to calculate the composite index using the principal component method by Nichkasova et al. (2020).

The second methodological approach (qualitative) measures the level of financial development of countries based on the qualitative characteristics of the financial sector. Such qualitative characteristics of financial development are the depth, accessibility, and efficiency of the financial market and the sector of financial organizations. This approach is supported by Cihak et al. (2012), Sahay et al. (2015), Ito and Kawai (2018), Naceur et al. (2019), Canh et al. (2020), M. A. Khan et al. (2020), Kavya and Santhakumar (2020),

Abdmoulah (2021), Gnangnon (2021), Canh and Thanh (2020), Canh and Thanh (2021), Dutta and Meierrieks (2021), and others.

Some publications suggest using other qualitative characteristics, such as the stability of the financial sector (Sanfilippo-Azofra et al. 2018).

Within the framework of the second methodological approach, the system of indicators includes three levels: upper (general), medium (qualitative characteristics), and lower (quantitative characteristics). At the same time, indicators of a higher order are calculated by aggregating indicators of a lower level.

A typical list of indicators of the middle and lower levels is given in Table 2.

**Table 2.** Indicators of the quality of financial development of countries.

| Indicators | Financial Markets | Financial Institutions |
|---|---|---|
| Depth | 1. Ratio of total value of traded shares to GDP, % <br> 2. Ratio of international debt securities of the government to GDP, % <br> 3. Ratio of total debt securities of financial corporations to GDP, % <br> 4. Ratio of total debt securities of non-financial corporations to GDP, % | 1. Ratio of loans to the private sector to GDP, % <br> 2. Ratio of pension funds' assets to GDP, % <br> 3. Ratio of UIF assets to GDP, % <br> 4. Ratio of insurance premiums to GDP, % |
| Accessibility | 1. Market capitalization outside the top 10 largest companies, % <br> 2. Total number of debt issuers (domestic and foreign, financial and non-financial corporations) per 100,000 adults | 1. Bank branches per 100,000 adults <br> 2. ATMs per 100,000 adults |
| Efficiency | Stock market turnover ratio (trading volume/stock market capitalization) | 1. Net interest margin of banks, % <br> 2. Interest rate spread, % <br> 3. Ratio of non-interest income to total income, % <br> 4. Overheads to total assets ratio, % <br> 5. Return on assets, % <br> 6. Return on equity, % |
| Stability | | Z-score of banks |

Source: Authors' research based on Cihak et al. (2012); Sahay et al. (2015); Ito and Kawai (2018); Naceur et al. (2019); Canh et al. (2020); M. A. Khan et al. (2020); Kavya and Santhakumar (2020); Abdmoulah (2021); Gnangnon (2021); Canh and Thanh (2020); Canh and Thanh (2021); and Dutta and Meierrieks (2021).

The second methodological approach looks more structured than the first approach. In addition, it uses a larger number of primary indicators that reflect not only the scale of financial development but also characterize the availability of financial services, the efficiency, and stability of the financial sector. This approach to measuring financial development partially solves the problem of the versatility of this concept. At the same time, an increase in the number of primary indicators for measuring financial development does not lead to more objective, reasonable and reliable results of scientific research. This is explained by the fact that in practice, researchers use not all, but only some of these indicators. Different sets of these indicators can lead to opposite conclusions. For example, Moyo and Le Roux (2021) concluded that financial development indicators have a significant impact on economic growth. Cave et al. (2020) found a strong negative relationship between the development of the banking sector and economic growth, and a positive relationship between the development of the stock market and economic growth (up to a threshold, after which the effect becomes negative).

The results of the literature analysis can be summarized as follows:

1.　Methodological approaches use the same primary indicators and differ only in their grouping criteria (by markets and financial organizations—in the first approach and additionally by qualitative characteristics—in the second approach);

2.　Primary indicators are calculated mainly in relation to the gross domestic product (GDP). Gross domestic product is a measure widely used in macroeconomic research; however, it has certain disadvantages. In particular, the methods for calculating this indicator change periodically in different countries. At the same time, the adjustment of methods does not occur simultaneously in different countries and not in the same way. This makes the time series over a long period within the same country incomparable and reduces the objectivity of the results of cross-country analysis;

3.　Primary indicators do not cover all segments (instruments) of the financial market. In particular, indicators for such segments (instruments) of financial markets as monetary gold and special drawing rights (SDRs), insurance pension and standard guarantees, pension payments, derivative financial instruments, and employee stock options are not used;

4.　Primary indicators are not calculated for all sectors of the economy, but only for the sector of financial corporations. Moreover, financial flows cover all sectors of the economy without exception. The lack of analysis of the financial development of non-financial sectors of the economy, taking into account their relationship with the financial sector, makes the results of the analysis incomplete and insufficiently formative;

5.　In scientific research, as a rule, the analysis of the financial development of countries is carried out using not all of the primary indicators presented in Tables 1 and 2, but only some of them. At the same time, the set of indicators used differs from study to study. This leads to conflicting conclusions from the results of research.

All this indicates the need to search for new, more informative indicators of financial development.

## 3. Methodology

The hypothesis of this study is that that the matrix system of indicators formed on the basis of the financial balance sheet of the System of National Accounts (SNA 2008) and reflecting transactions in financial assets by financial instruments and economic sectors is an objective and easy-to-use tool to assess the level of financial development of countries. Its use makes it possible to increase the complexity and quality of the analysis of the financial development of countries, and it also forms a platform for making evidence-based and effective decisions in the development of state strategic documents.

The practical implementation of this idea requires: an initial matrix of financial assets; a matrix system of indicators for the quantitative assessment of the financial development of countries; criteria for the qualitative assessment of the financial development of countries; algorithms for the structural assessment of financial development, allowing to identify promising areas of financial development for individual countries; and testing of the developed indicators, criteria, and algorithms.

The study uses a structured system approach to the development of indicators and criteria for financial development and the methodology of System of National Accounts (SNA 2008), in particular, the account No. 620, Financial accounts—non-consolidated—SNA 2008 (OECD 2021).

The study is based on the methods of structured system, comparative, matrix, and gap analysis. Methods of structured system analysis were used to develop a matrix system of financial development indicators that reflect all financial flows and their interrelationships in national economies. The methods of comparative analysis were used to build country ratings by level of financial development, to develop criteria for assessing the international position of countries, as well as to analyze the structure of the financial development of countries in the context of financial instruments and sectors of the economy. Matrix methods and gap analysis methods were used to build a matrix of structural deviations

that allows the identification of promising directions of financial development for an individual country.

The study uses statistics from 39 countries that are current and potential members of OECD (OECD+). The list of analyzed countries, and the period of analysis, were formed by taking into account the availability and representativeness of official data, as well as the availability of matching units of measurement of these data. The statistics for the empirical study (financial assets) were taken from account No. 620, Financial accounts—non-consolidated—SNA 2008 for the period 2018–2019 (OECD 2021).

### 3.1. Matrix Indicator System

The proposed system of indicators is formed based on the matrix *A*. The elements of the matrix $a_{ij}$ are financial assets classified according to the types of financial instruments (index *i*) and sectors of the economy (index *j*) (Figure 1).

$$A = (a_{ij}) \tag{1}$$

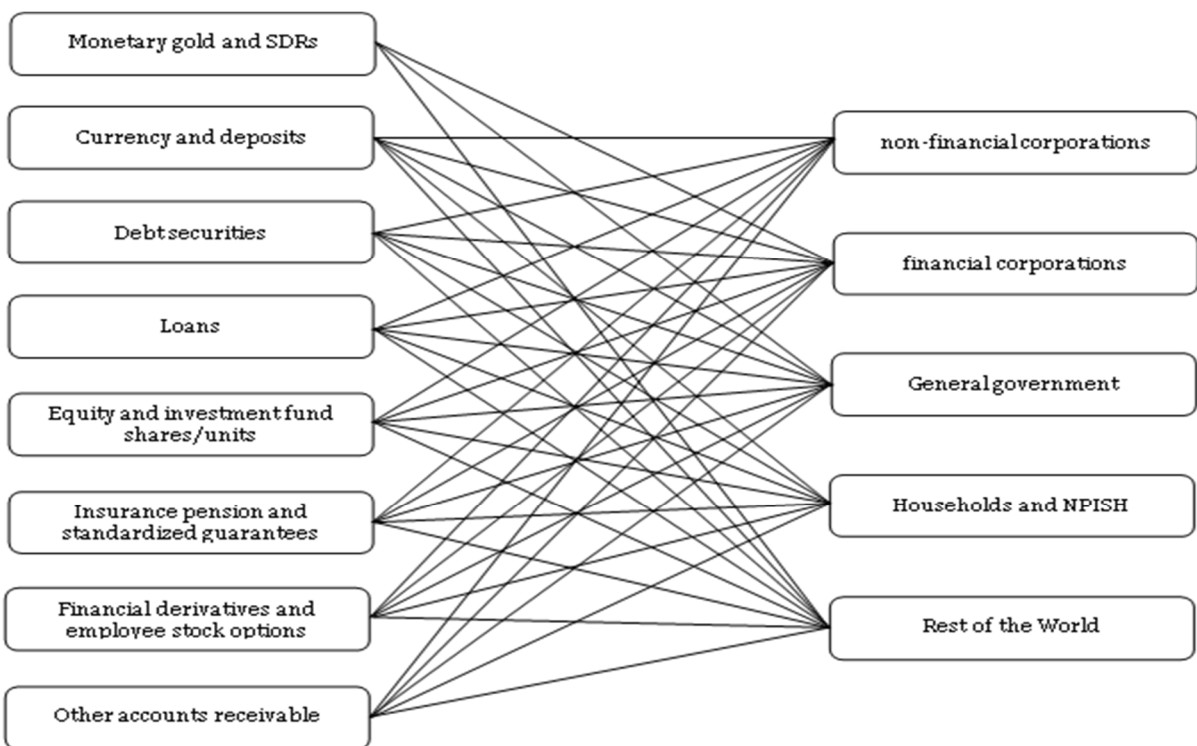

**Figure 1.** Classification and relationship of financial instruments and sectors of the economy. Source: authoring based on System of National Accounts (SNA 2008).

It is expedient to use the data of the financial balance of the SNA as the initial data for constructing matrix *A*. According to the System of National Accounts (SNA 2008), the matrix *A* has 8 × 5 dimensions with 8 financial assets and 5 economy sectors, as listed in Table 3.

**Table 3.** Matrix elements (A).

| Sectors (j)/Instruments (i) | Non-Financial Corporations (1) | Financial Corporations (2) | General Government (3) | Households and NPISH (4) | Rest of the World (5) |
|---|---|---|---|---|---|
| Monetary gold and SDRs (1) | $a_{11}$ | $a_{12}$ | $a_{13}$ | $a_{14}$ | $a_{15}$ |
| Currency and deposits (2) | $a_{21}$ | $a_{22}$ | $a_{23}$ | $a_{24}$ | $a_{25}$ |
| Debt securities (3) | $a_{31}$ | $a_{32}$ | $a_{33}$ | $a_{34}$ | $a_{35}$ |
| Loans (4) | $a_{41}$ | $a_{42}$ | $a_{43}$ | $a_{44}$ | $a_{45}$ |
| Equity and investment fund shares/units (5) | $a_{51}$ | $a_{52}$ | $a_{53}$ | $a_{54}$ | $a_{55}$ |
| Insurance pension and standardized guarantees (6) | $a_{61}$ | $a_{62}$ | $a_{63}$ | $a_{64}$ | $a_{65}$ |
| Financial derivatives and employee stock options (7) | $a_{71}$ | $a_{72}$ | $a_{73}$ | $a_{74}$ | $a_{75}$ |
| Other accounts receivable (8) | $a_{81}$ | $a_{82}$ | $a_{83}$ | $a_{84}$ | $a_{85}$ |

Source: authoring based on System of National Accounts (SNA 2008).

### 3.2. Quantifying the Level of Financial Development of Countries

To quantify the overall level of financial development of countries (*ID*), the elements of the matrix *A* must be summed up to a single value (*E*), and the resulting value should be divided by the population. In general, the level of financial development (*ID*) calculation formula will look like this:

$$E = \sum_{i=1}^{8} \sum_{j=1}^{5} a_{ij} \tag{2}$$

$$ID = E/P \tag{3}$$

where *P*—population.

The levels of development of individual segments of the financial sector (financial system) can be calculated based on two vector matrices (*ID$_i$*, *ID$_j$*) using the formulas:

$$ID_i = (\sum_{j=1}^{5} a_{ij})/P, \, i = 1, 8 \tag{4}$$

$$ID_j = (\sum_{i=1}^{8} a_{ij})/P, \, j = 1, 5 \tag{5}$$

### 3.3. Qualitative Assessment of the Level of Financial Development of Countries

For a qualitative description of financial development, it is advisable to use a special scale: "Criteria for the formation of the international rating of countries by the level of financial development" (Table 4). The rating system features five types of international scores: high, above average, average, below average, and low.

**Table 4.** Criteria for the formation of the international position of countries.

| International Positions | Criteria for the Formation of an International Position |
|---|---|
| High | $k_4 \leq ID$ |
| Above average | $k_3 \leq ID < k_4$ |
| Average | $k_2 \leq ID < k_3$ |
| Below average | $k_1 \leq ID < k_2$ |
| Low | $0 \leq ID < k_1$ |

Source: authoring.

Table 4 shows that the international positions of countries in terms of the level of financial development depend on the criteria corresponding to the values k1–k4. The proposed criteria should be formed based on the ranking of countries by *ID* indicators, followed by a breakdown of the total range of values into ranges that differ from each other by the order of values. It should be noted that the *ID* indicator characterizes the general level of financial development and includes individual indicators ($ID_i$, $ID_j$). At the same time, the quantitative values of the criteria for the indicators ($ID$, $ID_i$, $ID_j$) will be different.

*3.4. Structural Assessment of the Level of Financial Development of Countries*

A structural assessment of the financial development of countries consists of analyzing the current state of their financial sectors based on a matrix of transformed indicators (*D*):

The economic meaning of the transformation of indicators is to determine the specific weight of each element of matrix *A*. The share of each element will reveal the structural features of the financial sectors (financial systems) of countries, and through comparative analysis, determine promising areas of financial development.

The transformation of indicators is carried out by dividing the elements of matrix *A* by *E*:

$$D = A/E \tag{6}$$

To identify promising areas for the financial development of a particular country, it is advisable to use the formula:

$$IDd = D_e - D_t \tag{7}$$

where:

*IDd*—matrix of structural deviations;
$D_e$—matrix of reference values of transformed indicators;
$D_t$—matrix of actual values of the transformed indicators of the country, for which perspective directions of financial development are determined.

The relevance of this formula is determined by the fact that a higher level of financial development, as a rule, corresponds to a better structure of financial assets. In addition, according to Setiawan et al. (2021), countries with lower levels of financial development can emulate the growth principles of countries with higher levels of financial development.

It is recommended to use the values of the nearest best range (the structure of the elements of the nearest best international position) as a benchmark for countries scoring "above average", "average", "below average", "low". For countries with a high international score, such a benchmark will include the values of the country that occupies the first position. For a leader-country, it is advisable to develop new benchmarks to maintain leadership in the medium and long term.

Based on the results of the analysis, it is necessary to determine the maximum value of the gap for all elements of the *IDd* matrix. These maximum values should be considered as "points of growth" or the main directions of financial development. The implementation of these promising areas will best improve the international position of the country in the medium and long term.

## 4. Results

Testing of the developed matrix system of indicators of financial development was carried out in relation to current and potential members of the OECD (OECD+). The list of these countries is shown in Figure 2 and includes 39 countries. The statistics for the empirical study (financial assets) were taken from account No. 620, Financial accounts—non-consolidated—SNA (2008), for the period 2018–2019 (OECD 2021).

Using Formula (2), the level of financial development of the analyzed countries in 2018 and 2019 was calculated. Based on the results obtained, a rating of countries by the level of financial development was developed (Figure 2). Analysis of the ranking of countries shows the following. First, the list of top five countries in terms of financial development in 2018 included (in USD thousand) Luxembourg (49,796), Ireland (3186), Switzerland

(1872), the Netherlands (1619), and Denmark (1018). The same countries retained their leadership in 2019 (Luxembourg—49,763, Ireland—3539, Switzerland—2041, Netherlands—1668, Denmark—1115). These countries scored highly because they all belong to the group of economically developed countries and have international financial centers. Secondly, the lowest rating of countries in terms of financial development in 2018–2019 was received by Russia, Mexico, Colombia, Turkey, and India. All of these countries belong to the group of emerging economies and do not have international financial centers. Thirdly, the level of financial development of the leading countries is about a thousand times higher than the level of financial development of the countries got the lowest scores.

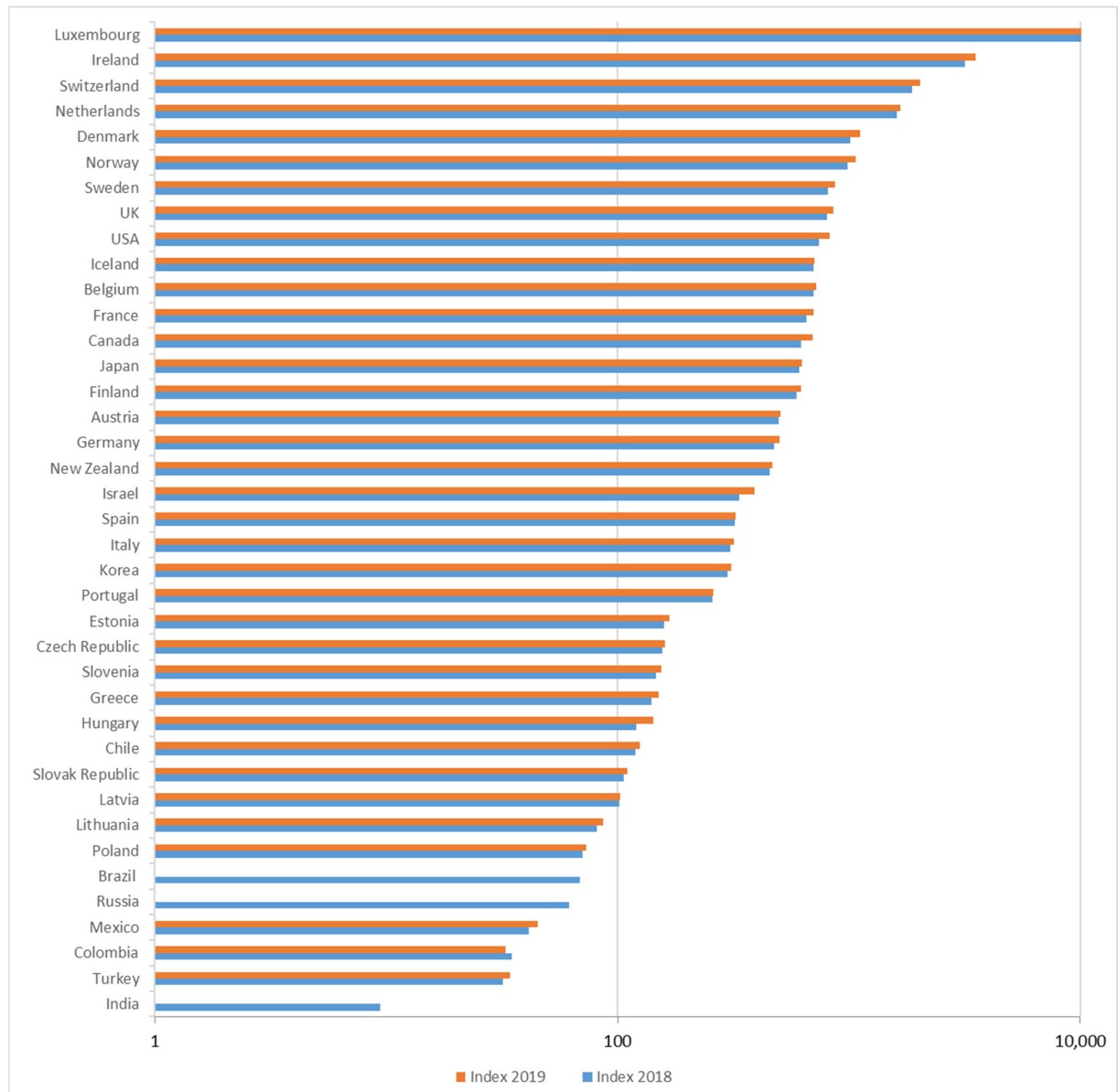

**Figure 2.** Rating of OECD+ countries by the level of financial development, USD thousands. Source: authors' calculations based on the official statistical data (OECD 2021).

To qualitatively characterize financial development, a special scale, "Criteria for the formation of the international position of OECD+ countries", was developed (Table 5). The levels of international ratings were formed using the rule of successive doubling of criteria values. This rule is used for the first time, and it allows, firstly, to take into account the spread in the values of the general indicator of financial development of OECD+ countries

(about 1000 times), and secondly, to ensure the relative uniformity of the distribution of countries by levels of financial development (types of international position).

**Table 5.** Qualitative scale for assessing the overall level of financial development of OECD+ countries.

| International Positions | Criteria for the Formation of an International Position, US$ Thousands |
|---|---|
| High | $1000 \leq ID$ |
| Above average | $500 \leq ID < 1000$ |
| Average | $250 \leq ID < 500$ |
| Below average | $125 \leq ID < 250$ |
| Low | $0 \leq ID < 125$ |

Source: authoring.

Based on the developed criteria, all analyzed countries were divided into groups in accordance with their international rating. The results of this distribution are presented in Table 6. The analysis of the obtained results shows that in 2018, five countries had a high international rating (Luxembourg, Ireland, Switzerland, Netherlands, and Denmark); "above average" was reached by ten countries (Norway, Sweden, UK, USA, Iceland, Belgium, France, Canada, Japan, and Finland); "average" scores went to eight countries (Austria, Germany, New Zealand, Israel, Spain, Italy, Korea, and Portugal); "below-average" went to eight countries (Estonia, Czech Republic, Slovenia, Greece, Hungary, Chile, Slovak Republic, and Latvia); and eight countries were ranked "low" (Lithuania, Poland, Brazil, Russia, Mexico, Colombia, Turkey, and India). In 2019, some countries managed to move to a new, higher level of financial development. For example, Norway moved up from "above average" to "high", whereas Austria and Germany moved from "average" to "above average".

**Table 6.** International positions of OECD+ countries, USD thousands.

| International Ranking | 2018 | 2019 |
|---|---|---|
| High | Luxembourg (49,796), Ireland (3186), Switzerland (1872), Netherlands (1619), Denmark (1018) | Luxembourg (49,763), Ireland (3539), Switzerland (2041), Netherlands (1668) Denmark (1115), Norway (1068) |
| Above average | Norway (990), Sweden (812), UK (808), USA (742), Iceland (705), Belgium (705) France (658), Canada (623), Japan (612), Finland (592) | Sweden (875), UK (855), USA (828), Iceland (712), Belgium (723) France (704), Canada (700), Japan (628), Finland (622), Austria (505), Germany (501) |
| Average | Austria (496), Germany (474) New Zealand (453), Israel (335), Spain (321), Italy (308), Korea (298), Portugal (256) | New Zealand (469), Israel (392), Spain (326), Italy (320), Korea (311), Portugal (258) |
| Below average | Estonia (159), Czech Republic (157), Slovenia (147), Greece (141) | Estonia (168), Czech Republic (160), Slovenia (154), Greece (150), Hungary (142) |
| Low | Hungary (121), Chile (120), Slovak Republic (107), Latvia (101), Lithuania (81), Poland (70), Brazil (69), Russia (62), Mexico (41), Colombia (35, Turkey (32), India (9) | Chile (124), Slovak Republic (110), Latvia (103), Lithuania (87), Poland (73), Brazil (*) Russia (*), Mexico (45), Colombia (33), Turkey (34), India (*) |

Source: authors' calculations based on the official statistical data (OECD 2021). * Note: No data available.

Based on Formula (5), a structural analysis of the financial development of the OECD+ countries was carried out in the context of types of financial assets (instruments) and sectors of the economy. The results of this analysis are presented in Figures 3 and 4.

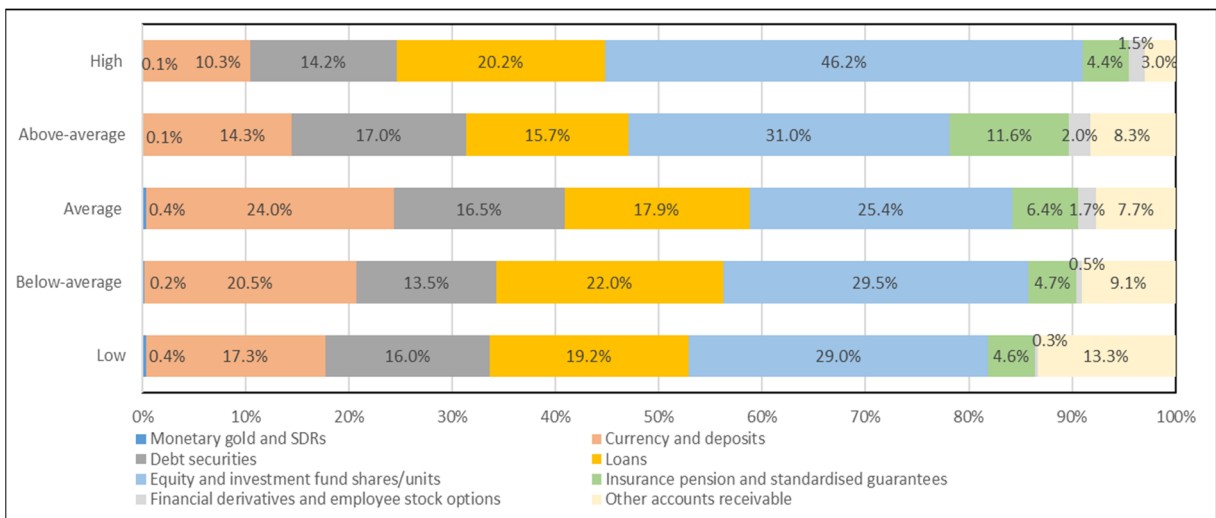

**Figure 3.** Instrumental structure of the financial assets of OECD+ countries in the context of international positions, 2019, %. Source: authors' calculations based on the official statistical data (OECD 2021). Note: the values of "Monetary gold and SDRs" are not visible in the figure due to their small value, which is less than 0.4%.

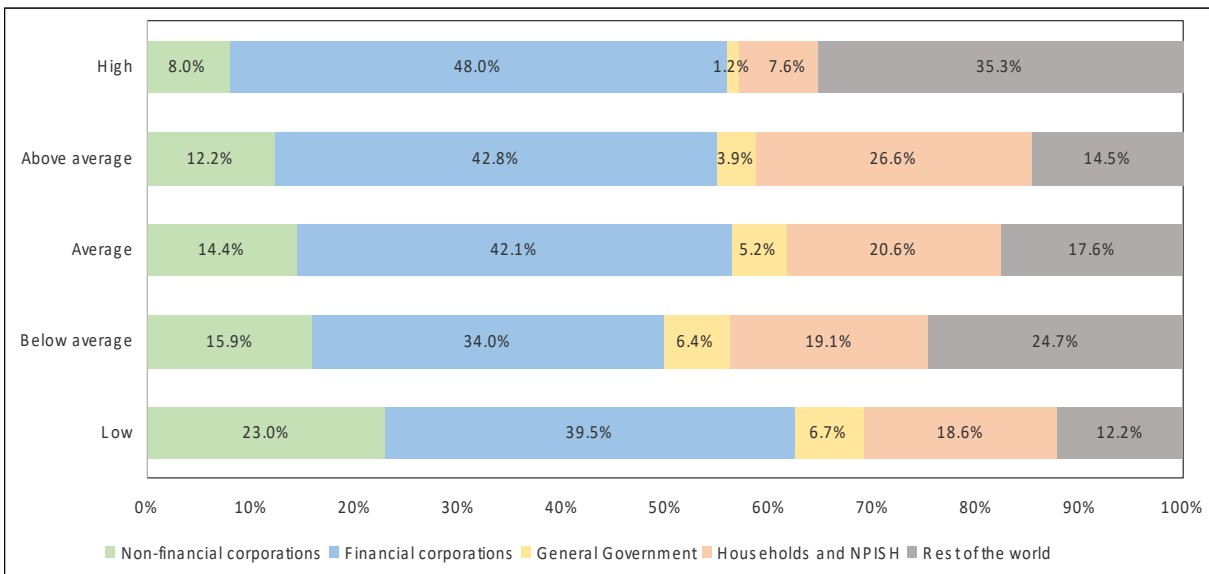

**Figure 4.** Sector structure of the financial assets of OECD+ countries in the context of international positions, 2019, %. Source: authors' calculations based on the official statistical data (OECD 2021).

Based on the analysis, the following main statements can be made. First, all analyzed countries have a different structure of financial assets, which differs both in terms of the scale of financial instruments used and in the volume of financial assets formed by individual sectors of the economy. At the same time, the structure of financial assets of countries with the same international position has fewer differences than countries with different international positions. Thirdly, each type of international position has its structural features. These features are as follows:

1.   A high international position is characterized by the maximum share of equity and investment fund shares/unit in combination with the maximum share of the financial

corporation sector and the rest of the world sector in terms of financial assets. This is due to the high level of development of the financial sector in economically developed countries, as well as a high level of confidence in their financial institutions and instruments;

2. The above average position is characterized by the maximum share of debt securities, insurance pension and standardized guarantees, as well as financial derivatives and employee stock options, combined with the maximum share of the households and NPISH sector. This confirms the emergence of modern financial instruments (debt, insurance, and derivatives) and the large role of households in financial development. In order to move to a higher level of financial development, these countries need to concentrate their efforts on the development of investment instruments and increase their reliability, which will increase the attractiveness of the financial market for foreign investment;

3. Countries with an average level of financial development are characterized by the maximum share of currency and deposits. Consequently, in these countries, the use of modern financial instruments is not yet sufficiently adopted, but favorable conditions have already been created for the formation of a sufficiently high level of confidence in the financial system. A further step to increase the level of financial development of these countries may be the development of the insurance and stock markets;

4. The "below average" score is characterized by the maximum share of loans in financial assets. This is due to the insufficient level of development of the stock market, and the fact that the banking system in these countries occupies a dominant position in their financial system. In this regard, these countries need to develop stock market instruments;

5. A low position is characterized by the maximum share of other accounts receivable in financial instruments and the maximum share of non-financial corporations and general government sectors among other groups of countries. This means that for countries of this group, the financial sector is much less developed than in more economically developed countries. And the maximum share of other accounts receivable means a lower level of use of the main financial instruments. Consequently, the next step in increasing the financial development of these countries will be to both increase the range of financial instruments and further develop the financial sector of the economy.

In general, the structural features of the financial development of countries occupying different international positions make it possible to consider the structural transformation of financial assets as a guideline for the successful financial development of individual countries, and on this basis, to determine promising directions for their further development.

The identification of promising areas for the financial development of a particular country was carried out in relation to Italy, which in 2019 occupied an "average" international position in terms of the level of financial development. For this purpose, based on formula 6, the following were calculated:

1. Matrix of reference values of Italy's transformed indicators (transformed indicators of the closest best range—"above average"). The results of the calculations are presented in Table 7;

2. Matrix of actual values of the converted indicators of Italy (Table 8).

3. Matrix of structural deviations of the actual values of the transformed indicators of Italy from the reference values (Table 9).

**Table 7.** Reference values of indicators (d ij) e, for Italy, 2019, %.

| Financial Instruments | Economic Sectors | | | | |
|---|---|---|---|---|---|
| | Non-Financial Corporations | Financial Corporations | General Government | Households and NPISH | Rest of the World |
| Monetary gold and SDRs | 0.00 | 0.08 | 0.02 | 0.00 | 0.02 |
| Currency and deposits | 1.65 | 4.78 | 0.50 | 5.55 | 2.07 |
| Debt securities | 0.19 | 10.22 | 0.57 | 1.17 | 4.44 |
| Loans | 0.98 | 11.75 | 0.63 | 0.20 | 1.78 |
| Equity and investment fund shares/units | 5.09 | 10.48 | 0.90 | 10.27 | 5.69 |
| Insurance pension and standardized guarantees | 0.13 | 1.67 | 0.00 | 8.95 | 0.05 |
| Financial derivatives and employee stock options | 0.02 | 1.36 | 0.00 | 0.00 | 0.83 |
| Other accounts receivable | 4.22 | 2.17 | 0.89 | 0.32 | 0.37 |

Source: authors' calculations based on the official statistical data (OECD 2021).

**Table 8.** Actual indicator values (d ij) t Italy, 2019, %.

| Financial Instruments | Economic Sectors | | | | |
|---|---|---|---|---|---|
| | Non-Financial Corporations | Financial Corporations | General Government | Households and NPISH | Rest of the World |
| Monetary gold and SDRs | 0.00 | 0.66 | 0.00 | 0.00 | 0.0 |
| Currency and deposits | 2.2 | 8.20 | 0.54 | 8.52 | 4.54 |
| Debt securities | 0.33 | 12.58 | 0.25 | 1.56 | 6.30 |
| Loans | 0.48 | 12.24 | 0.89 | 0.07 | 1.48 |
| Equity and investment fund shares/units | 4.53 | 6.00 | 1.01 | 9.76 | 3.68 |
| Insurance pension and standardized guarantees | 0.07 | 0.09 | 0.01 | 6.54 | 0.01 |
| Financial derivatives and employee stock options | 0.09 | 0.89 | 0.00 | 0.00 | 0.61 |
| Other accounts receivable | 3.43 | 0.16 | 0.72 | 0.82 | 0.64 |

Source: authors' calculations based on the official statistical data (OECD 2021).

**Table 9.** Structural Variance Matrix (IDd) for Italy, 2019, p.p.

| Financial Instruments | Economic Sectors | | | | |
|---|---|---|---|---|---|
| | Non-Financial Corporations | Financial Corporations | General Government | Households and NPISH | Rest of the World |
| Monetary gold and SDRs | 0.00 | −0.58 | 0.02 | 0.00 | −0.03 |
| Currency and deposits | −0.61 | −3.42 | −0.04 | −2.97 | −2.47 |
| Debt securities | −0.14 | −2.36 | 0.32 | −0.39 | −1.84 |
| Loans | 0.50 | −0.49 | −0.26 | 0.13 | 0.30 |
| Equity and investment fund shares/units | 0.56 | 4.48 | −0.11 | 0.51 | 2.01 |
| Insurance pension and standardized guarantees | 0.06 | 1.58 | −0.01 | 2.41 | 0.04 |
| Financial derivatives and employee stock options | −0.07 | 0.47 | 0.00 | 0.00 | 0.22 |
| Other accounts receivable | 0.79 | 2.02 | 0.17 | −0.50 | −0.27 |

Note: ▢—positive values of deviations (>0), ▢—no deviations (=0), ▢—insignificant negative values of deviations (equal and less than 1.0 p.p.), ▢—significant negative values of deviations (more than 1.0 p.p.). Source: authors' calculations based on the official statistical data (OECD 2021).

Furthermore, all obtained structural deviations were ranked according to their magnitude as follows: positive values of deviations (>0), no deviations (=0), insignificant

negative values of deviations (equal to and less than 1.0 p.p.), significant negative values of deviations (more than 1.0 p.p.). The most significant values of deviations (more than 1.0 percentage points) were taken as the priority directions for Italy's financial development. They included the growth of financial assets in the form of currency and deposits of financial corporations, households, NPISH, and rest of the world, as well as the debt securities of financial corporations and the rest of the world.

The results of testing of the developed matrix system of indicators of financial development confirmed that this system of indicators is an objective and easy-to-use tool for assessing the level of financial development of countries. Its use makes it possible to increase the complexity and quality of the analysis of the financial development of countries, and it also forms a platform for making evidence-based and effective decisions in the development of state strategic documents.

## 5. Discussion

As a result of the study, new indicators have been proposed to measure the financial development of countries. These indicators are calculated based on data from the financial balances of the SNA. The financial balance data includes transactions in financial assets and financial liabilities. The study used data on financial assets that characterize the directions of use of financial resources. The analysis of financial development based on data on financial liabilities would provide no less interesting and useful results. In addition, it is proposed for determining the new indicators in relation to the population, and not to GDP. This is due to the fact that the population, in comparison with GDP, is an indicator that is more resistant to crisis phenomena and does not contain methodological discrepancies in its calculations. At the same time, this indicator can be refined by not using the entire population, but its economically active part.

A unique feature of the study is the focus on the structural analysis of the financial development of countries, which makes it possible to identify promising areas of financial development. At the same time, quite interesting results, in our opinion, can be obtained based on the analysis of the time series of countries' financial assets. This direction is promising from the point of view of further research.

In the study, the values of the criteria for the formation of international positions were calculated in relation to one group of countries—OECD+ countries. To use the new system of indicators on a global scale, the calculation of the criteria needs to be clarified. Doing this requires an analysis of financial assets for all countries of the world or a representative sample of these countries.

The study proposed criteria that make it possible to determine the promising directions of the financial development of countries based on the maximum structural deviations from the reference values. At the same time, the structure of the elements of the nearest best international position or the leading country in the ranking of countries (if the analyzed country occupies a high international position) should be used as a reference. For a country that is a leader in the ranking of countries, the proposal is to develop new benchmarks to maintain leadership in the medium and long term. The solution to this problem was beyond the scope of the present study. In this regard, the development of these reference values is a promising direction for further scientific research.

Another important point in the study is that the new system of indicators was tested in relation to the general indicator of the financial development of countries. The analysis of financial development based on particular indicators (indicators of the development of financial instruments and sectors of the economy) could be a good addition to the study and could provide a clearer picture of the formation of financial assets and their management at the level of national jurisdictions.

The study confirmed the results of previous studies (Naceur et al. 2019), which concluded that the level of financial market development in European countries with economies in transition is rather low. By diversifying and developing the structure of local financial markets and institutions, it would be possible to use national resources for the country

more efficiently. This can be done by creating and developing an appropriate debt securities market for short and long-term investments (Kreso and Begovic 2013). In addition, the results of the study confirm the conclusion of Setiawan et al. (2021) that developing countries can emulate the growth principles of developed countries in financial market development. At the same time, the results showed that international financial centers are an attribute of the financial sector of countries with a high international position in terms of financial development. This conclusion is inconsistent with the conclusion of previous studies, according to which globalization helps to mobilize economic growth, but does not contribute to financial development (Kandil et al. 2015).

## 6. Conclusions

In recent years, there has been a significant increase in the number of strategic documents aimed at the financial development of countries. This has created the challenge of improving the quality of measurement of financial development. The present study is aimed at solving this problem. The aim of the study is to develop a new system of indicators and criteria for measuring the level of financial development of countries, which make it possible to assess the scale of financial development, range countries by levels of development, and identify priority areas for this development.

As a result of the study, a matrix system of indicators was developed to measure the financial development of countries. This system of indicators is based on the matrix of financial assets of countries, which characterizes the relationships between of all types of financial assets (instruments and sectors of the economy). Simultaneous recording of financial transactions across the entire range of financial instruments and sectors of the economy, as well as their interrelations, is a relatively new direction in measuring financial development.

This is confirmed by comparing the proposed matrix system of indicators with the existing indicators of financial development. For example, in the study of Ito and Kawai (2018), a list of the following indicators was used for a comprehensive assessment of the financial development of the country: the ratio of domestic credit to the banking sector to GDP, %; the ratio of capitalization of the stock market to GDP, %; the ratio of the volume of traded shares and bonds to GDP, %; the ratio of the external debt of the state to GDP, %; and the ratio of the volume of insurance premiums to GDP, %. Based on the results of the analysis of these indicators, two main conclusions can be drawn. First, the indicators reviewed by Ito and Kawai (2018) do not cover all financial assets, but only some of them. Second, Ito and Kawai (2018) calculate indicators in relation to GDP. In this study, the matrix system of indicators contains a complete list of financial assets, and indicators are calculated in relation to the population.

In the course of the study, the level of financial development of 39 countries (actual and potential OECD members) in 2018 and 2019 was calculated. On the basis of the results obtained, the rating of countries by the level of financial development was built. The rating analysis showed that in 2018, the top five leaders in terms of financial development included Luxembourg, Ireland, Switzerland, the Netherlands, and Denmark. The same countries maintained their leadership in 2019. The bottom lines of the rating were occupied by Russia, Mexico, Colombia, Turkey, India. At the same time, the level of financial development of the leading countries was about a thousand times higher than the level of financial development of the countries that were at the bottom of the rating.

The analysis of the results also showed that in 2018, 5 countries had a high international position (Luxembourg, Ireland, Switzerland, the Netherlands, and Denmark; the position was "above average" for 10 countries (Norway, Sweden, UK, USA, Iceland, Belgium, France, Canada, Japan, and Finland); the "average" position was held by 8 countries (Austria, Germany, New Zealand, Israel, Spain, Italy, Korea, and Portugal); the position "below average" was held by 4 countries (Estonia, Czech Republic, Slovenia, and Greece), and the "low" position was held by 12 countries (Hungary, Chile, Slovak Republic, Latvia, Lithuania, Poland, Brazil, Russia, Mexico, Colombia, Turkey, and India). In 2019, some

countries managed to move to a new, higher level of financial development. For example, Norway moved up from "above average" to "high", whereas Austria and Germany moved from "average" to "above average".

A comprehensive analysis of the financial development of OECD+ countries confirmed that these countries have a different instrumental and sectoral structure of financial assets. An analysis of the structural deviation matrix for Italy identified priority areas for the country's further financial development. They were the growth of financial assets in the form of currency and deposits of financial corporations, households, NPISH, and rest of the world, as well as the debt securities of financial corporations and the rest of the world.

The limitations of the study are as follows: 1. the new system of indicators for financial development uses data on the financial assets of countries, which characterize financial flows in terms of the allocation of financial resources; 2. the study assumes that the population indicator of countries is a more objective indicator than GDP since it is more resistant to crisis phenomena and does not contain methodological discrepancies in its calculations; and 3. the study is focused on the analysis of the structural features of the financial development of countries at different levels of financial development, which made it possible to consider structural differences as priority areas for further development.

The proposed system of indicators for measuring the financial development of countries can be the subject of further research in terms of using data on financial liabilities of the financial balances of the SNA. Studies of time series of data to assess the value of financial assets and financial liabilities would, in turn, allow for a more objective assessment of the cyclical nature of financial development and identification of factors that have a significant impact on the level of development and stability of financial systems. The generalization of the results of these studies could lead to the formation of uniform standards and requirements for the development of strategic documents on financial development, taking into account the sustainability of the processes of this development.

**Author Contributions:** Conceptualization, methodology, formal analysis, investigation writing—review and editing, G.G., E.Z.; project administration, G.G. All authors have read and agreed to the published version of the manuscript.

**Funding:** This research was funded by the Ministry of Education and Science of the Russian Federation (Priority 2030—Strategic Academic Leadership Program), grant number H-481-99_2021-2022 (2021–2022).

**Institutional Review Board Statement:** Not applicable.

**Informed Consent Statement:** Not applicable.

**Data Availability Statement:** The data will be made available on request from the corresponding author.

**Conflicts of Interest:** The authors declare no conflict of interest.

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
