# Peer review of "Assessment of Financial Development of Countries Based on the Matrix of Financial Assets"

_economies, doi:10.3390/economies10050122_

Round 1

Reviewer 1 Report

Comments on the Paper "New Indicator System for Financial Development Assessment of Countries"

I read with pleasure this paper. It is well-written, and I believe that it could contribute to the relevant literature.

I would like to focus on the theoretical part of the analysis. First, I would suggest that the authors further elaborate on the differences between their proposed multidimensional indicator of financial development and the indicators of quality and quantity of financial development proposed by Ito and Kawai (2018) (well cited by the authors in the paper).

Reference

Ito, H., and Kawai, M., 2018. Quantity and Quality Measures of Financial Development: Implications for Macroeconomic Performance. Policy Research Institute, Ministry of Finance, Japan, Public Policy Review 14(5): 803-833.

Author Response

Response to Reviewer 1 Comments

Point 1: I would like to focus on the theoretical part of the analysis. First, I would suggest that the authors further elaborate on the differences between their proposed multidimensional indicator of financial development and the indicators of quality and quantity of financial development proposed by Ito and Kawai (2018) (well cited by the authors in the paper).

Referenc: Ito, H., and Kawai, M., 2018. Quantity and Quality Measures of Financial Development: Implications for Macroeconomic Performance. Policy Research Institute, Ministry of Finance, Japan, Public Policy Review 14(5): 803-833.

Response 1: We took into account this remark and supplemented the section "Conclusions".

Reviewer 2 Report

the present manuscript describes a possible indicator system for analyse the countries' financial development.

  • Moderate English changes required
  • Abstract: the reviewer suggests write again this part of the manuscript without headings
  • The reviewer should consider to have a more careful attention to the reference style
  • The reviewer suggests taking a look at the Figure and Table sources indicating in a more detailed way the sources themselves
  • Method is not  adequately described and it is necessary to write again this part of the manuscript
  • the results are not clearly presented and it is necessary to write again this part of the manuscript

Author Response

Response to Reviewer 2 Comments

Point 1: Moderate English changes required.

Response 1: We took this comment into account and corrected the text.

Point 2: Abstract: the reviewer suggests write again this part of the manuscript without headings.

Response 2: We took this comment into account and corrected the text.

Point 3: The reviewer should consider to have a more careful attention to the reference style.

Response 3: We took this comment into account and corrected the text.

Point 4: The reviewer suggests taking a look at the Figure and Table sources indicating in a more detailed way the sources themselves.

Response 4: We took this comment into account and corrected the text.

Point 5: Method is not  adequately described and it is necessary to write again this part of the manuscript.

Response 5: We took this comment into account and corrected the text.

Point 6: The results are not clearly presented and it is necessary to write again this part of the manuscript.

Response 6: We took this comment into account and corrected the text.

Reviewer 3 Report

The study is interesting, and the outcomes of this paper could contribute to the advancement of knowledge. In general, this article is written clearly but it requires some revisions.

For more details, see the comments below.

Author Response

Response to Reviewer 3 Comments

Point 1: Deletion of the landmarks that underlie the construction of the abstract (context, methods, results and conclusions); all these sections must be joined so that the final text to be coherent.

Response 1: We took this comment into account and corrected the text.

Point 2:  The research problem stated by the authors (“lack of formalized strategic goals ... in the strategic documents ... and the incompatibility of the pathways for achieving these goals”) is too general and intuitive; a clear and objective reformulation of the research issue is needed.

Response 2: We took this comment into account and corrected the text.

Point 3: Regarding the “methods”, are required additions concerning the sample, processed data and period.

Response 3: We took this comment into account and corrected the text.

Point 4: In the results section, the recommendation to avoid exaggeration must be followed; after all, the authors use well-known indicators; the novelty is given by the way in which they are used /combined to obtain more representative results.

Response 4: We took this comment into account and corrected the text.

Point 5:  Regarding to the conclusions drawn in the abstract, the authors point out that the value of the article is given by the “proposed methodology” (which reinforces the above); in the light of the last two recommendations, the title of the article should also be re-evaluated.

Response 5: We took into account this remark and corrected the title of the article.

Point 6: One of the publisher's recommendations is to keep the introduction easy to understand for scientists working outside the subject of the paper.

Response 6: We took this comment into account and corrected the text.

Point 7: Regarding the idea formulated in the introduction („A significant increase in the number of strategic documents, as well as the absence of quantitatively expressed strategic goals...”), the authors are asked to provide more details to facilitate understanding.

Response 7: We took this comment into account and corrected the text.

Point 8: Authors are invited to review citations. Some of the articles mentioned in the introduction differ from those recorded in the references; for example: Moyo & Le Roux (2021 or 2020); Wang, Wen & Xu (2018 or 2017). The second citation also appears in the literature review.

Response 8: We took this comment into account and corrected the text.

Point 9: The indicators must be presented correctly and in a uniform manner (in the current version “share” and “%” are used alternatively; authors must use internationally recognized terms.

Response 9: We took this comment into account and corrected the text.

Point 10: The “real% rate” indicator must comply with the previously formulated recommendation.

Response 10: We took this comment into account and corrected the text.

Point 11: The indicators presented in table 2 can also be found in the World Bank database; however, the authors state as the source – “authoring”; authors are invited to re-evaluate all tables and assess whether they are proposing new indicators or taking over and processing from existing databases (to be cited).

Response 11: We took this comment into account and corrected the text.

Point 12: Abbreviations must be detailed (e.g., SDRs).

Response 12: We took this comment into account and corrected the text.

Point 13: In section 3.4, the first paragraph ends with “(D):”.

Response 13: We took this comment into account and corrected the text.

Point 14: For the identification of areas with opportunities for financial development, formula (7) is recommended; the authors are asked to detail what this recommendation is based on.

Response 14: We took this comment into account and corrected the text.

Point 15: Table 5 states that the thresholds are set using the rule of successive doubling of values; in order to comply with this rule, the “low” threshold should be between 0 and 125 (and "Below-average" should be between 125 and 250); even if the rule of setting thresholds is justifiable, the authors are asked to point out whether this rule has been used before.

Response 16: We took this comment into account and corrected the text.

Point 16:  If there is no argument for maintaining the current thresholds (0-100 and 100-250), then Table 5 needs to be revised (some of the countries listed in the “Below-average” category should move to the “Low” category.

Response 16: We took this comment into account and corrected the text.

Point 17:  Iin figure 3, the values for the variable “Monetary gold and SDRs” are missing.

Response 17: We took this comment into account and corrected the text.

Point 18: In describing the structural features of international positions (pages 12-13), the authors formulate ideas for the 5 positions; the intention of the authors was to highlight the “instrumental structure of financial assets” by capturing the maximum values; authors are asked to re-evaluate point five (“maximum share of Monetary gold and SDRs” or minimum share of Monetary gold and SDRs).

Response 18: We took this comment into account and corrected the text.

Point 19: The authors discuss the results and how they can be interpreted in the light of previous studies. The limitations and future directions of research are highlighted. In this section, the authors note that: “The study confirmed the results of previous studies, which concluded that the level of financial market development in European countries with economies in transition is rather low.” Related to this idea, the authors are asked to cite the sources that confirm the results obtained.

Response 19: We took this comment into account and corrected the text.

Point 20: Conclusions. Authors should review the first three sentences in this section because they are identical to the first 3 sentences in the abstract. The first sentence also appears in the beginning of the introduction.  The authors are invited to make this review in the light of the recommendations given in this review letter.

Response 20: We took this comment into account and corrected the text.

Point 21: References. In the references are presented articles that are not cited in the paper (e.g., 14, 29, 56). The authors are invited to detail the extent to which paragraphs 40 and 41 (from the references) correspond to Nguyen et al. (2020a); Nguyen et al. (2020 б) of the text. Cyrillic letters need to be replaced.

Response 21: We took this comment into account and corrected the text.

Round 2

Reviewer 2 Report

The new version of the present manuscript is well written and present an interesting topic

Author Response

We have taken into account the reviewer's response.

Reviewer 3 Report

I have read the new version of the manuscript. The incorporated changes reveal a qualitative improvement of the study.

At this time, only a few minor revisions are still needed.

1. Abbreviations must be defined the first time they appear. See M2 and GDP.

2. In the first review, I have recommended that the authors review the first paragraph of the "Conclusions" section (because it repeated some of the text from the abstract). Given that the authors have completely redone the abstract, the first paragraph of the conclusions may remain.

Author Response

Response to Reviewer 3 Comments

Point 1: Abbreviations must be defined the first time they appear. See M2 and GDP.

Response 1: We took this comment into account and corrected the text.

Point 2:  In the first review, I have recommended that the authors review the first paragraph of the "Conclusions" section (because it repeated some of the text from the abstract). Given that the authors have completely redone the abstract, the first paragraph of the conclusions may remain.

Response 2: We took this comment into account and corrected the text.